# Faster Measurement for Formaldehyde Emissions from Veneered Particleboard Based on the Standardized Desiccator Method

Jijuan Zhang [1,2] , Zhaozhen Gui [1], Yang Chen [1], Li Xue [1], Feifei Song [2] and Zhongfeng Zhang [1,2,*]

1    College of Furniture and Art Design, Central South University of Forestry and Technology, Changsha 410004, China
2    Green Furniture Engineering Technology Research Center, National Forestry and Grassland Administration, Changsha 410004, China
*    Correspondence: t19990735@csuft.edu.cn

**Abstract:** Desiccator method is a fast and effective way to measure formaldehyde emissions from wood-based panels. This method is popular in the Chinese furniture industry and testing institutions. It is also an important method for production control due to its characteristics of low cost, fast speed, and simple operation. In order to further increase the measurement speed of the desiccator method, this study focuses on the impact of temperature and time conditions in regard to standard GB/T 17657-2013. The corresponding relationships for the measurement of formaldehyde emissions between the standardized desiccator method and those under different temperature and time conditions were studied. Four different experimental conditions were used: 60 °C for 6 h, 40 °C for 6 h, 43 °C for 4 h, and 45 °C for 4 h. The results showed that under 40 °C for 6 h the formaldehyde emissions measured using desiccator method were about twice as much as those under 20 °C for 24 h, at a correlation coefficient of R = 0.820. Under 45 °C for 4 h, the formaldehyde emissions measured using desiccator method were almost near equilibrium to emissions under 20 °C for 24 h, which was supported by a correlation coefficient of R = 0.955. A corresponding relationship between the formaldehyde emissions measurement results under these two conditions and those under the standardized conditions was observed. This relationship can be applied in the actual production control in the furniture industry in order to shorten the formaldehyde measurement time from 24 h to 6 h and 4 h, which can greatly improve measurement efficiency.

**Keywords:** desiccator method; formaldehyde emissions measurement; temperature; veneered particleboard; wood-based panels

## 1. Introduction

Formaldehyde has been classified as a potentially dangerous carcinogen and an important environmental pollutant by the World Health Organization and the United States Environmental Protection Agency. Formaldehyde can cause many health damaging effects, such as acute toxicity, oxidative stress and inflammation, genotoxicity, neurotoxicity, cardiovascular effects, etc. It can also induce leukemia, nasopharyngeal carcinoma, sinus, and other cancers. National Toxicology Program (NTP) and International Agency for Research on Cancer (IARC) have listed formaldehyde as a "confirmed carcinogen" [1–3]. Formaldehyde emissions of wood-based panels mainly includes the residual free formaldehyde in the panel and the formaldehyde diffusion due to the adhesive degradation during the use of the panel; the content of free formaldehyde is the most prominent in the adhesives used. Chinese standard GB18580-2017 [4] stipulates that the measurement of formaldehyde emissions from wood-based panels shall be carried out in accordance with the 1 m$^3$ climate chamber method in the GB/T 17657-2013 standard [5]; the desiccator method, and the perforator method were used for production quality control in enterprise. The formaldehyde emissions standards of wood-based panels in different countries are shown in Table 1 [6].

**Table 1.** Formaldehyde emissions standards for wood-based panels in Europe, USA, Japan, Australia, and China.

| Country | Standard | Test Method | Board Class | Limit Value |
|---|---|---|---|---|
| Europe | EN13986: 2005 | Perforator EN ISO 12460-5 | E1-unfaced particleboard, MDF/HDF, OSB | $\leq$8 mg/100g * |
| | | Chamber EN 717-1 | E1-particleboard, MDF/HDF, OSB | $\leq$0.1 ppm ** |
| | | Gas analysis EN 717-2 | E1-unfaced plywood, solid wood panels, laminated veneer lumber (LVL) | $\leq$3.5 mg/m$^2$ h |
| | | Gas analysis EN 717-2 | E1-coated, overlaid, or veneered particleboard, OSB, fiberboard, plywood, solid wood panels, LVL, cement-bonded particleboard | $\leq$3.5 mg/m$^2$ h |
| USA | ANSI A 208.1 & 2 | ASTM E1333 (chamber) | Particleboard/MDF | $\leq$0.18 or 0.09 ppm/ $\leq$0.21 or 0.11 ppm |
| Japan | JIS A 5908 (2015) and 5905 | JIS A 1460 (Desiccator) | F**/F***(E0)/F****(SE0) | $\leq$1.5 mg/L/$\leq$0.5 mg/L/ $\leq$0.3 mg/L |
| Australia and New Zealand | AS/NZS 1859/1 (2017) and 2 | AS/NZS 4266.16 (Desiccator) | E0-particleboard, MDF/E1-particleboard /E1/MDF | $\leq$0.5 mg/L/$\leq$1.5 mg/L/ $\leq$1.0 mg/L |
| China | GB18580-2017 | GB/T 17657-2013 (chamber) | E1-MDF, particleboard, plywood, LVL, or veneered wood-based panel | $\leq$0.124 mg/m$^3$ |

* E3 30–60 mg/100 g, E2 8–30 mg/100 g, E1 5–8 mg/100 g, E0 $\leq$3 mg/100 g, super E0 $\leq$1.5 mg/100 g. ** 0.05 ppm boards can be marked with an environmental label ("Blue Angel"), 0.03 ppm boards are about equal to the Japanese emission class F****.

The test time for the climate chamber method is long, which is not conducive to the production quality control needed for industrial application. The desiccator method is popular in the Chinese furniture industry and testing institutions. This method is an important method for industrial production control with the characteristics of low cost, fast speed, and simple operation. The main factors affecting the measurement of formaldehyde emissions using desiccator method are container cleanliness, tightness, specimen balance, temperature, and time (Meyer et al., 1983, Rybicky et al., 1983) [7,8]. Measuring formaldehyde emissions from wood-based panels can be a complicated process, which can be affected by (1) factors related to the materials, such as type of panel, wood species, adhesive, and overlay used for the panels; (2) factors related to the environment, such as temperature, humidity, air velocity, and air exchange rate; (3) factors related to treatment; (4) factors related to panel fabrication process, such as resin content, moisture content of the panel, and others [9]. Yin (2020) researched veneered particleboard by investigating the influences of water absorption and panel's loading rate on the determination of formaldehyde emissions in panels by desiccator method. Absorbed water was positively correlated with formaldehyde emissions, while carrying capacity was negatively correlated with formaldehyde emissions [10].

Kim S. and Kim H. (2005) analyzed the effect of various indoor temperatures on the formaldehyde emissions from building finishing material. The flooring materials were exposed to temperatures of 37 °C and 50 °C while furniture materials were only exposed to room temperature. The results show that after bake-out the formaldehyde emissions of flooring materials were much lower than those of furniture materials. This proved that the temperature factor should be considered for the management of indoor air quality [11]. Lin C. (2009) found that the formaldehyde emission rate and its concentration increased 1.5–12.9 times when the temperature was raised from 15 °C to 30 °C [12]. Chi D. (2014) tested the formaldehyde emissions of plywood, medium-density fiberboard (MDF), block board, and laminate at different temperatures and loading rates using a 1 m$^3$ small chamber.

The results showed that the higher temperature accelerated the formaldehyde release; the higher the temperature, the faster the initial emission rate, and the greater the final concentration. It was found that the formaldehyde emissions would increase 10%–30% if the temperature was increased by 5 °C [13]. These experimental studies confirmed the positive effect of temperature on formaldehyde release because temperature increased the kinetic energy and sped up the diffusion rate of formaldehyde molecules, and the high temperature led to decomposition of the adhesives, which increased the formaldehyde release. However, these methods cannot estimate the emissions under other temperatures. According to Mayers (1985), the effect of temperature on indoor formaldehyde concentration showed an exponential relationship. The diffusion coefficient (D), partition coefficient (K), and the initial emittable concentration (Cm, 0) were the three key parameters used to predict the formaldehyde emissions [14]. Zhang Y. (2007) used the C-history method to measure the diffusion coefficient (D) and the partition coefficient (K) of formaldehyde in dry building materials (particleboard, vinyl floor, medium-density fiberboard, and high-density fiberboard) at temperatures of 18, 30, 40, and 50 °C. The results showed that temperature has significant effect on both the partition coefficient (K) and the diffusion coefficient (D) of formaldehyde emissions from the four materials tested, the partition coefficients (K) decreased while the diffusion coefficient (D) increased with the increase in temperature [15].

Si L. (2014) selected three factors to study the measurement of formaldehyde emissions from wood-based panels using the desiccator method: test temperature, reaction temperature of absorption solution, and reaction time of absorption solution. The test temperature significantly affected the measured value of formaldehyde emissions, while the other two factors had no significant impact [16]. Liu Y. (2019) stressed that the temperature and humidity must be kept constant in the measurement of formaldehyde emissions when using desiccator method; different temperatures have different effects on the data. The cured urea formaldehyde resin released more formaldehyde when the temperature exceeded 30 °C [17]. Shi J. (2018) studied the relationship between the test results of desiccator method under different temperature conditions; the formaldehyde emissions from the panel increased with the increase in temperature [18].

To further increase the measurement speed of the desiccator method, this study focuses on the change in temperature and time conditions of the desiccator method in the GB/T 17657-2013 standard. The corresponding relationships for the measurement of formaldehyde emissions between the standardized desiccator method and those under different temperature and time conditions in order to shorten the formaldehyde measurement time of the desiccator method for it to be applied in industrial production control.

## 2. Materials and Methods

### 2.1. Methods

Standardized desiccator method: The test was carried out in accordance with the desiccator specified in standard GB/T 17657-2013 "Test methods of evaluating the properties of wood-based panels and surface decorated wood-based panels". At a temperature of 20 °C, the test sample of a certain surface area was placed in the desiccator. The formaldehyde released by the test sample was absorbed by a certain volume of water, and the formaldehyde content in water was measured throughout 24 h.

Different temperature and time conditions of the desiccator method: According to previous studies, the test temperature can accelerate the release of formaldehyde [11–15]. By increasing the temperature and shortening the time to improve the measurement speed of desiccator method, the conditions were set as: 60 °C for 6 h, 40 °C for 6 h, 43 °C for 4 h, and 45 °C for 4 h.

### 2.2. Materials

The test material was a three-layer veneered particleboard with a thickness of 18 mm and density of 0.63 g/cm$^3$, which is the largest consumption in the panel furniture industry. In order to make the test results more representative, the particleboard samples

were made of three adhesives: urea-formaldehyde (UF), phenol-formaldehyde (PF), and diphenylmethane diisocyanate (MDI). The length of test samples was (150 $\pm$ 1.0) mm, and the width was (50 $\pm$ 1.0) mm. The total surface area of the test sample (including side, both ends, and surface) was close to 1800 cm$^2$, and the number of test samples were determined accordingly.

Two groups of samples were selected from the same panel and recorded as samples A and B. Group A was tested according to the setting temperature and time conditions, and group B was tested according to the 20 °C for 24 h conditions, which is the standardized desiccator method specified in standard GB/T 17657-2013. The formaldehyde emissions of the two groups of the same plate were compared.

The following reagents were used: Acetyl acetone (analytical purity), ammonium acetate (analytical purity), glacial acetic acid (analytical purity), and 10.1% formaldehyde standard solution (CH$_2$O).

The software used to perform the data analysis of the results was IBM SPSS Statistics 25.

## 3. Results and Discussion

*3.1. Formaldehyde Emissions Measured by the Desiccator at 60 °C for 6 h and 20 °C for 24 h*

In order to test the effect of the significant increase in temperature on the formaldehyde emissions, the conditions of group A were set as collection at 60 °C for 6 h, and those of group B were set as collection at 20 °C for 24 h (Table 2).

**Table 2.** Formaldehyde emissions data measured by the desiccator at 60 °C for 6 h and 20 °C for 24 h.

| Group A (60 °C & 6 h) | | | Group B (20 °C & 24 h) | | |
|---|---|---|---|---|---|
| No. | As * | FE ** (mg/L) | No. | As | FE (mg/L) |
| 1 | 0.764 | 5.73 | 1 | 0.076 | 0.53 |
| 2 | 1.400 | 10.60 | 2 | 0.061 | 0.41 |
| 3 | 1.187 | 8.98 | 3 | 0.084 | 0.60 |
| 4 | 1.182 | 8.93 | 4 | 0.037 | 0.24 |
| 5 | 0.992 | 7.45 | 5 | 0.091 | 0.64 |
| 6 | 1.286 | 9.75 | 6 | 0.061 | 0.42 |

* As: absorbance. ** FE: formaldehyde emissions.

By comparing the formaldehyde emissions under the two conditions, it was observed that the increase in temperature had a significant impact on formaldehyde emissions, and the formaldehyde emissions at 60 °C were more than ten times of that at 20 °C, which was consistent with the conclusion of Liu (2019): the cured urea formaldehyde resin releases more formaldehyde when the temperature exceeds 30 °C [17]. Shi J. (2018) observed that when the temperature reaches 80 °C, the formaldehyde emissions exceed the limit value by more than twice [18]. Yang Y. (2016) found that a 5 °C increase in temperature could increase the emissions by 1.3–2.5 times [19]. Qiu x. (2020) found that the formaldehyde emissions from laminate flooring and solid wood composite floor at 30 °C was 1.9 times and 1.5 times higher than that at 23 °C, respectively [20].

The test results of this stage show that, at a high temperature, the collected formaldehyde emissions are far more than that under normal temperature. However, the formaldehyde emission trends under these two conditions were different; there was no significant regularity, which is not suitable for further research. However, these results confirmed that high temperatures affect the formaldehyde emissions, and it provides a reference for future tests.

*3.2. Formaldehyde Emissions Measured by the Desiccator at 40 °C for 6 h and 20 °C for 24 h*

Considering the normal working environment of particleboard, the maximum indoor temperature in summer is about 40 °C. The conditions of group A were set as a collection at 40 °C for 6 h, and those of group B were also set as a collection at 20 °C for 24 h (Table 3).

**Table 3.** Formaldehyde emissions data measured by the desiccator at 40 °C for 6 h and 20 °C for 24 h.

| Group A (40 °C & 6 h) | | | Group B (20 °C & 24 h) | | | Group A (40 °C & 6 h) | | | Group B (20 °C & 24 h) | | |
|---|---|---|---|---|---|---|---|---|---|---|---|
| No. | As | FE (mg/L) | No. | As | FE (mg/L) | No. | As | FE (mg/L) | No. | As | FE (mg/L) |
| 1 | 0.117 | 0.80 | 1 | 0.066 | 0.46 | 28 | 0.069 | 0.44 | 28 | 0.077 | 0.52 |
| 2 | 0.240 | 1.76 | 2 | 0.113 | 0.83 | 29 | 0.107 | 0.73 | 29 | 0.066 | 0.43 |
| 3 | 0.087 | 0.56 | 3 | 0.046 | 0.31 | 30 | 0.181 | 1.30 | 30 | 0.089 | 0.64 |
| 4 | 0.122 | 0.83 | 4 | 0.088 | 0.62 | 31 | 0.094 | 0.61 | 31 | 0.054 | 0.37 |
| 5 | 0.086 | 0.56 | 5 | 0.061 | 0.42 | 32 | 0.14 | 0.97 | 32 | 0.099 | 0.71 |
| 6 | 0.207 | 1.50 | 6 | 0.135 | 1.00 | 33 | 0.192 | 1.38 | 33 | 0.123 | 0.91 |
| 7 | 0.148 | 1.05 | 7 | 0.062 | 0.41 | 34 | 0.105 | 0.69 | 34 | 0.077 | 0.51 |
| 8 | 0.106 | 0.71 | 8 | 0.068 | 0.45 | 35 | 0.166 | 1.17 | 35 | 0.152 | 1.09 |
| 9 | 0.076 | 0.50 | 9 | 0.054 | 0.35 | 36 | 0.081 | 0.54 | 36 | 0.055 | 0.33 |
| 10 | 0.067 | 0.43 | 10 | 0.039 | 0.23 | 37 | 0.179 | 1.20 | 37 | 0.088 | 0.59 |
| 11 | 0.085 | 0.57 | 11 | 0.040 | 0.24 | 38 | 0.144 | 1.02 | 38 | 0.083 | 0.57 |
| 12 | 0.081 | 0.53 | 12 | 0.055 | 0.35 | 39 | 0.135 | 0.94 | 39 | 0.078 | 0.51 |
| 13 | 0.027 | 0.11 | 13 | 0.014 | 0.04 | 40 | 0.024 | 0.09 | 40 | 0.018 | 0.04 |
| 14 | 0.184 | 1.33 | 14 | 0.062 | 0.41 | 41 | 0.146 | 0.93 | 41 | 0.091 | 0.52 |
| 15 | 0.134 | 0.93 | 15 | 0.031 | 0.18 | 42 | 0.102 | 0.69 | 42 | 0.051 | 0.34 |
| 16 | 0.093 | 0.61 | 16 | 0.091 | 0.65 | 43 | 0.098 | 0.66 | 43 | 0.034 | 0.21 |
| 17 | 0.122 | 0.84 | 17 | 0.032 | 0.19 | 44 | 0.105 | 0.70 | 44 | 0.045 | 0.30 |
| 18 | 0.099 | 0.68 | 18 | 0.089 | 0.63 | 45 | 0.074 | 0.47 | 45 | 0.039 | 0.25 |
| 19 | 0.075 | 0.47 | 19 | 0.029 | 0.16 | 46 | 0.042 | 0.22 | 46 | 0.016 | 0.07 |
| 20 | 0.024 | 0.08 | 20 | 0.011 | 0.03 | 47 | 0.128 | 0.85 | 47 | 0.057 | 0.39 |
| 21 | 0.138 | 0.93 | 21 | 0.057 | 0.38 | 48 | 0.104 | 0.66 | 48 | 0.041 | 0.28 |
| 22 | 0.020 | 0.05 | 22 | 0.017 | 0.05 | 49 | 0.113 | 0.73 | 49 | 0.061 | 0.41 |
| 23 | 0.218 | 1.60 | 23 | 0.136 | 0.99 | 50 | 0.087 | 0.56 | 50 | 0.056 | 0.39 |
| 24 | 0.158 | 1.17 | 24 | 0.090 | 0.64 | 51 | 0.117 | 0.79 | 51 | 0.041 | 0.26 |
| 25 | 0.168 | 1.19 | 25 | 0.098 | 0.66 | 52 | 0.146 | 0.93 | 52 | 0.072 | 0.52 |
| 26 | 0.102 | 0.70 | 26 | 0.063 | 0.41 | 53 | 0.167 | 1.10 | 53 | 0.088 | 0.65 |
| 27 | 0.181 | 1.32 | 27 | 0.127 | 0.90 | | | | | | |

It was observed that the trend in formaldehyde emissions measured using the desiccator method under 40 °C for 6 h had a similar reading to those under 20 °C for 24 h, and there was a preliminary multiple relationship. Through statistics, it was found that there was a double relationship in 38 of the 53 groups of the experimental data. The data that did not conform to the multiple relationships showed that the formaldehyde emissions measured using the desiccator method under the condition of 40 °C for 6 h was greater than that under 20 °C for 24 h.

Through the correlation analysis of the 53 groups of data, the formaldehyde emissions measured using the desiccator method at 20 °C for 24 h was taken as the *Y*-axis, and the formaldehyde emissions measured using the desiccator method at 40 °C for 6 h was taken as the *X*-axis. The corresponding relationship between two different conditions were plotted and possessed a correlation of $R^2 = 0.6722$ (Figure 1). Through a correlation analysis using SPSS, the correlation coefficient of the results between the two conditions were determined to be $R = 0.820$, indicating that the correlation was strong. A significance test was carried out; the results were extremely significant, sig.(2-tailed) $p = 0.00 < 0.01$.

According to the significance test and correlation analysis, there was a significant linear correlation between this group of experiments data (Table 4).

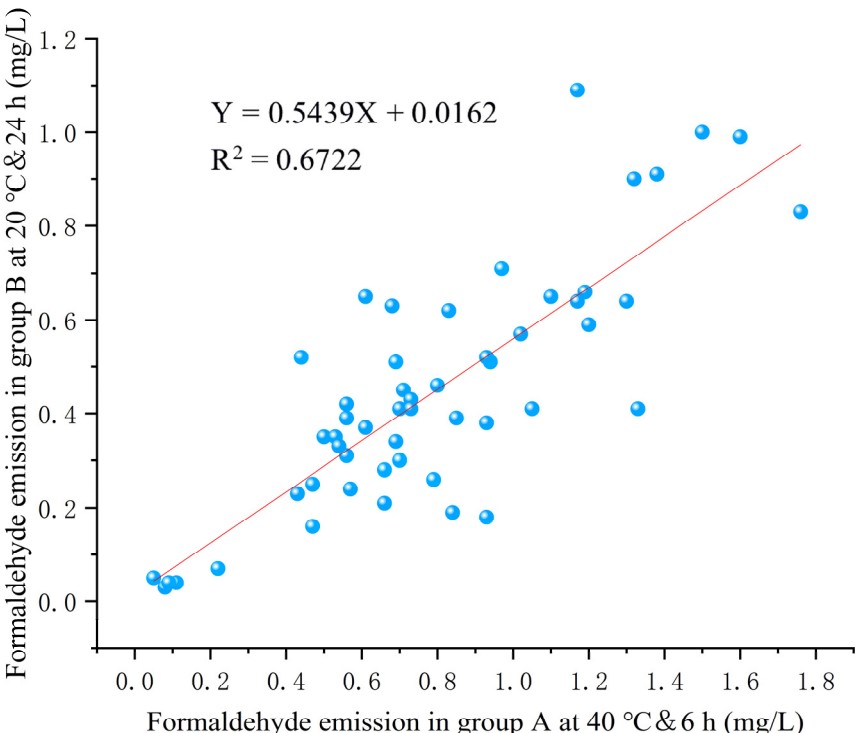

**Figure 1.** Correlation analysis scatter plot of formaldehyde emissions at 40 °C for 6 h and 20 °C for 24 h.

**Table 4.** Correlation analysis of formaldehyde emissions at 40 °C for 6 h and 20 °C for 24 h.

|  |  | 40 °C & 6 h | 20 °C & 24 h |
|---|---|---|---|
|  | Pearson correlation | 1 | 0.820 ** |
| 40 °C & 6 h | Sig. (2-tailed) |  | 0.000 |
|  | Number of cases | 53 | 53 |
|  | Pearson correlation | 0.820 ** | 1 |
| 20 °C & 24 h | Sig. (2-tailed) | 0.000 |  |
|  | Number of cases | 53 | 53 |

** At the 0.01 level (2-tailed), the correlation was significant.

Based on these results, industry can measure the formaldehyde emissions using the desiccator method at 40 °C for 6 h to preliminarily determine whether the formaldehyde emissions were qualified for production control emergency.

*3.3. Formaldehyde Emissions Measured by the Desiccator Method at 43 °C for 4 h and 20 °C for 24 h*

To further shorten the measurement time, the conditions of group A were set as collection at 43 °C for 4 h, and those of group B were set as collection at 20 °C for 24 h. The test results are shown in Table 5.

The test results indicated that there were too many abnormal data of formaldehyde emissions measured using the desiccator method at 43 °C for 4 h and 20 °C for 24 h, such as 1, 2, 7, 9, 18, and 20, which reflected that the formaldehyde emissions did not reach a stable state at 43 °C (Table 5). Therefore, it is impossible to objectively establish a relationship with the formaldehyde emissions measured using the desiccator method at 20 °C for 24 h.

**Table 5.** Formaldehyde emissions data measured by the desiccator at 43 °C for 4 h and 20 °C for 24 h.

| Group A (43 °C & 4 h) | | | Group B (20 °C & 24 h) | | | Group A (43 °C & 4 h) | | | Group B (20 °C & 24 h) | | |
|---|---|---|---|---|---|---|---|---|---|---|---|
| No. | As | FE (mg/L) | No. | As | FE (mg/L) | No. | As | FE (mg/L) | No. | As | FE (mg/L) |
| **1 *** | **0.068** | **0.40** | **1** | **0.081** | **0.50** | 13 | 0.063 | 0.37 | 13 | 0.066 | 0.42 |
| **2 *** | **0.069** | **0.41** | **2** | **0.097** | **0.61** | 14 | 0.067 | 0.40 | 14 | 0.059 | 0.37 |
| 3 | 0.02 | 0.04 | 3 | 0.019 | 0.07 | 15 | 0.019 | 0.04 | 15 | 0.016 | 0.04 |
| 4 | 0.092 | 0.58 | 4 | 0.09 | 0.59 | 16 | 0.02 | 0.05 | 16 | 0.015 | 0.04 |
| 5 | 0.015 | 0.01 | 5 | 0.014 | 0.03 | 17 | 0.072 | 0.44 | 17 | 0.053 | 0.32 |
| 6 | 0.09 | 0.62 | 6 | 0.085 | 0.56 | **18 *** | **0.106** | **0.69** | **18** | **0.083** | **0.55** |
| **7 *** | **0.08** | **0.54** | **7** | **0.104** | **0.70** | 19 | 0.074 | 0.45 | 19 | 0.066 | 0.42 |
| 8 | 0.077 | 0.52 | 8 | 0.091 | 0.61 | **20 *** | **0.08** | **0.51** | **20** | **0.098** | **0.65** |
| **9 *** | **0.094** | **0.65** | **9** | **0.082** | **0.54** | 21 | 0.071 | 0.45 | 21 | 0.065 | 0.40 |
| 10 | 0.092 | 0.63 | 10 | 0.093 | 0.63 | 22 | 0.018 | 0.05 | 22 | 0.022 | 0.08 |
| 11 | 0.068 | 0.44 | 11 | 0.06 | 0.38 | 23 | 0.064 | 0.40 | 23 | 0.058 | 0.35 |
| 12 | 0.086 | 0.57 | 12 | 0.081 | 0.53 | | | | | | |

* Abnormal data.

### 3.4. Formaldehyde Emissions Measured by the Desiccator Method at 45 °C for 4 h and 20 °C for 24 h

The conditions of group A were set as collection at 45 °C for 4 h, and those of group B were set as a collection at 20 °C for 24 h. The test results were shown in Table 6.

**Table 6.** Formaldehyde emissions data measured by the desiccator at 45 °C for 4 h and 20 °C for 24 h.

| Group A (45 °C & 4 h) | | | Group B (20 °C & 24 h) | | | Group A (45 °C & 4 h) | | | Group B (20 °C & 24 h) | | |
|---|---|---|---|---|---|---|---|---|---|---|---|
| No. | As | FE (mg/L) | No. | As | FE (mg/L) | No. | As | FE (mg/L) | No. | As | FE (mg/L) |
| 1 | 0.052 | 0.32 | 1 | 0.066 | 0.31 | 85 | 0.123 | 0.78 | 85 | 0.087 | 0.60 |
| 2 | 0.074 | 0.46 | 2 | 0.069 | 0.45 | 86 | 0.058 | 0.38 | 86 | 0.039 | 0.24 |
| 3 | 0.074 | 0.46 | 3 | 0.059 | 0.38 | 87 | 0.071 | 0.48 | 87 | 0.052 | 0.34 |
| 4 | 0.012 | 0.02 | 4 | 0.011 | 0.02 | 88 | 0.035 | 0.21 | 88 | 0.023 | 0.13 |
| 5 | 0.062 | 0.39 | 5 | 0.040 | 0.24 | 89 | 0.013 | 0.04 | 89 | 0.013 | 0.04 |
| 6 | 0.096 | 0.66 | 6 | 0.105 | 0.73 | 90 | 0.106 | 0.75 | 90 | 0.072 | 0.49 |
| 7 | 0.037 | 0.18 | 7 | 0.038 | 0.23 | 91 | 0.012 | 0.03 | 91 | 0.009 | 0.01 |
| 8 | 0.021 | 0.05 | 8 | 0.017 | 0.05 | 92 | 0.137 | 0.98 | 92 | 0.100 | 0.70 |
| 9 | 0.111 | 0.72 | 9 | 0.109 | 0.72 | 93 | 0.108 | 0.76 | 93 | 0.061 | 0.40 |
| 10 | 0.076 | 0.45 | 10 | 0.055 | 0.30 | 94 | 0.073 | 0.45 | 94 | 0.052 | 0.30 |
| 11 | 0.113 | 0.75 | 11 | 0.075 | 0.50 | 95 | 0.105 | 0.70 | 95 | 0.092 | 0.60 |
| 12 | 0.085 | 0.52 | 12 | 0.062 | 0.41 | 96 | 0.085 | 0.54 | 96 | 0.079 | 0.50 |
| 13 | 0.077 | 0.45 | 13 | 0.040 | 0.23 | 97 | 0.102 | 0.66 | 97 | 0.082 | 0.51 |
| 14 | 0.026 | 0.10 | 14 | 0.020 | 0.07 | 98 | 0.023 | 0.07 | 98 | 0.021 | 0.04 |
| 15 | 0.024 | 0.11 | 15 | 0.017 | 0.05 | 99 | 0.081 | 0.52 | 99 | 0.078 | 0.49 |
| 16 | 0.130 | 0.95 | 16 | 0.086 | 0.58 | 100 | 0.018 | 0.03 | 100 | 0.019 | 0.03 |
| 17 | 0.153 | 1.13 | 17 | 0.117 | 0.84 | 101 | 0.089 | 0.56 | 101 | 0.048 | 0.25 |
| 18 | 0.045 | 0.27 | 18 | 0.037 | 0.20 | 102 | 0.120 | 0.80 | 102 | 0.110 | 0.71 |
| 19 | 0.046 | 0.28 | 19 | 0.038 | 0.21 | 103 | 0.061 | 0.30 | 103 | 0.054 | 0.25 |
| 20 | 0.086 | 0.55 | 20 | 0.075 | 0.47 | 104 | 0.022 | 0.01 | 104 | 0.021 | 0.01 |
| 21 | 0.074 | 0.46 | 21 | 0.062 | 0.38 | 105 | 0.088 | 0.52 | 105 | 0.081 | 0.47 |
| 22 | 0.067 | 0.42 | 22 | 0.062 | 0.38 | 106 | 0.101 | 0.61 | 106 | 0.089 | 0.52 |
| 23 | 0.071 | 0.45 | 23 | 0.062 | 0.40 | 107 | 0.072 | 0.39 | 107 | 0.045 | 0.19 |
| 24 | 0.070 | 0.45 | 24 | 0.062 | 0.38 | 108 | 0.019 | 0.04 | 108 | 0.017 | 0.04 |
| 25 | 0.091 | 0.61 | 25 | 0.075 | 0.47 | 109 | 0.104 | 0.68 | 109 | 0.080 | 0.51 |
| 26 | 0.092 | 0.61 | 26 | 0.062 | 0.40 | 110 | 0.107 | 0.71 | 110 | 0.106 | 0.71 |
| 27 | 0.104 | 0.70 | 27 | 0.102 | 0.69 | 111 | 0.168 | 1.21 | 111 | 0.141 | 1.01 |
| 28 | 0.077 | 0.50 | 28 | 0.075 | 0.47 | 112 | 0.115 | 0.76 | 112 | 0.092 | 0.60 |
| 29 | 0.154 | 1.07 | 29 | 0.077 | 0.49 | 113 | 0.048 | 0.31 | 113 | 0.034 | 0.21 |
| 30 | 0.113 | 0.75 | 30 | 0.083 | 0.51 | 114 | 0.108 | 0.74 | 114 | 0.082 | 0.55 |
| 31 | 0.099 | 0.65 | 31 | 0.097 | 0.61 | 115 | 0.117 | 0.81 | 115 | 0.112 | 0.78 |
| 32 | 0.111 | 0.75 | 32 | 0.083 | 0.51 | 116 | 0.072 | 0.46 | 116 | 0.061 | 0.39 |

**Table 6.** *Cont.*

| Group A (45 °C & 4 h) | | | Group B (20 °C & 24 h) | | | Group A (45 °C & 4 h) | | | Group B (20 °C & 24 h) | | |
|---|---|---|---|---|---|---|---|---|---|---|---|
| No. | As | FE (mg/L) | No. | As | FE (mg/L) | No. | As | FE (mg/L) | No. | As | FE (mg/L) |
| 33 | 0.090 | 0.59 | 33 | 0.062 | 0.42 | 117 | 0.092 | 0.61 | 117 | 0.053 | 0.33 |
| 34 | 0.144 | 1.01 | 34 | 0.108 | 0.78 | 118 | 0.085 | 0.55 | 118 | 0.072 | 0.46 |
| 35 | 0.090 | 0.62 | 35 | 0.051 | 0.33 | 119 | 0.104 | 0.70 | 119 | 0.102 | 0.68 |
| 36 | 0.018 | 0.05 | 36 | 0.015 | 0.04 | 120 | 0.038 | 0.23 | 120 | 0.037 | 0.18 |
| 37 | 0.148 | 1.09 | 37 | 0.109 | 0.79 | 121 | 0.045 | 0.26 | 121 | 0.037 | 0.22 |
| 38 | 0.017 | 0.05 | 38 | 0.018 | 0.04 | 122 | 0.289 | 2.09 | 122 | 0.204 | 1.49 |
| 39 | 0.075 | 0.47 | 39 | 0.059 | 0.35 | 123 | 0.023 | 0.09 | 123 | 0.017 | 0.08 |
| 40 | 0.097 | 0.64 | 40 | 0.055 | 0.35 | 124 | 0.075 | 0.47 | 124 | 0.060 | 0.37 |
| 41 | 0.017 | 0.04 | 41 | 0.014 | 0.04 | 125 | 0.079 | 0.51 | 125 | 0.055 | 0.34 |
| 42 | 0.062 | 0.38 | 42 | 0.049 | 0.30 | 126 | 0.076 | 0.55 | 126 | 0.051 | 0.35 |
| 43 | 0.070 | 0.44 | 43 | 0.052 | 0.33 | 127 | 0.078 | 0.49 | 127 | 0.046 | 0.27 |
| 44 | 0.038 | 0.20 | 44 | 0.024 | 0.12 | 128 | 0.079 | 0.51 | 128 | 0.053 | 0.32 |
| 45 | 0.017 | 0.05 | 45 | 0.016 | 0.05 | 129 | 0.150 | 1.04 | 129 | 0.127 | 0.88 |
| 46 | 0.096 | 0.64 | 46 | 0.067 | 0.41 | 130 | 0.082 | 0.55 | 130 | 0.068 | 0.45 |
| 47 | 0.021 | 0.07 | 47 | 0.018 | 0.04 | 131 | 0.099 | 0.68 | 131 | 0.089 | 0.60 |
| 48 | 0.017 | 0.04 | 48 | 0.017 | 0.04 | 132 | 0.017 | 0.05 | 132 | 0.015 | 0.04 |
| 49 | 0.015 | 0.03 | 49 | 0.016 | 0.03 | 133 | 0.184 | 1.32 | 133 | 0.168 | 1.21 |
| 50 | 0.096 | 0.63 | 50 | 0.094 | 0.62 | 134 | 0.014 | 0.03 | 134 | 0.012 | 0.02 |
| 51 | 0.095 | 0.62 | 51 | 0.088 | 0.58 | 135 | 0.053 | 0.26 | 135 | 0.042 | 0.21 |
| 52 | 0.018 | 0.04 | 52 | 0.014 | 0.03 | 136 | 0.087 | 0.53 | 136 | 0.044 | 0.22 |
| 53 | 0.096 | 0.65 | 53 | 0.076 | 0.52 | 137 | 0.094 | 0.56 | 137 | 0.063 | 0.35 |
| 54 | 0.072 | 0.45 | 54 | 0.043 | 0.25 | 138 | 0.065 | 0.33 | 138 | 0.042 | 0.19 |
| 55 | 0.128 | 0.89 | 55 | 0.093 | 0.63 | 139 | 0.109 | 0.66 | 139 | 0.072 | 0.42 |
| 56 | 0.032 | 0.16 | 56 | 0.023 | 0.10 | 140 | 0.058 | 0.29 | 140 | 0.054 | 0.29 |
| 57 | 0.014 | 0.02 | 57 | 0.013 | 0.02 | 141 | 0.208 | 1.47 | 141 | 0.127 | 0.95 |
| 58 | 0.015 | 0.03 | 58 | 0.014 | 0.03 | 142 | 0.132 | 0.89 | 142 | 0.117 | 0.88 |
| 59 | 0.071 | 0.43 | 59 | 0.047 | 0.28 | 143 | 0.044 | 0.18 | 143 | 0.026 | 0.15 |
| 60 | 0.025 | 0.12 | 60 | 0.019 | 0.07 | 144 | 0.059 | 0.29 | 144 | 0.037 | 0.23 |
| 61 | 0.029 | 0.15 | 61 | 0.024 | 0.12 | 145 | 0.022 | 0.02 | 145 | 0.008 | 0.02 |
| 62 | 0.089 | 0.59 | 62 | 0.071 | 0.46 | 146 | 0.068 | 0.42 | 146 | 0.043 | 0.26 |
| 63 | 0.091 | 0.61 | 63 | 0.069 | 0.45 | 147 | 0.016 | 0.04 | 147 | 0.013 | 0.04 |
| 64 | 0.060 | 0.37 | 64 | 0.050 | 0.30 | 148 | 0.015 | 0.03 | 148 | 0.011 | 0.03 |
| 65 | 0.076 | 0.49 | 65 | 0.049 | 0.29 | 149 | 0.070 | 0.43 | 149 | 0.050 | 0.31 |
| 66 | 0.098 | 0.69 | 66 | 0.079 | 0.54 | 150 | 0.086 | 0.56 | 150 | 0.050 | 0.32 |
| 67 | 0.020 | 0.06 | 67 | 0.017 | 0.05 | 151 | 0.074 | 0.49 | 151 | 0.054 | 0.36 |
| 68 | 0.082 | 0.54 | 68 | 0.055 | 0.34 | 152 | 0.112 | 0.77 | 152 | 0.074 | 0.51 |
| 69 | 0.096 | 0.66 | 69 | 0.057 | 0.35 | 153 | 0.011 | 0.02 | 153 | 0.008 | 0.02 |
| 70 | 0.018 | 0.06 | 70 | 0.012 | 0.05 | 154 | 0.081 | 0.54 | 154 | 0.047 | 0.30 |
| 71 | 0.017 | 0.05 | 71 | 0.012 | 0.05 | 155 | 0.109 | 0.74 | 155 | 0.096 | 0.66 |
| 72 | 0.015 | 0.04 | 72 | 0.012 | 0.04 | 156 | 0.081 | 0.54 | 156 | 0.050 | 0.32 |
| 73 | 0.019 | 0.07 | 73 | 0.012 | 0.04 | 157 | 0.049 | 0.30 | 157 | 0.036 | 0.19 |
| 74 | 0.152 | 1.09 | 74 | 0.086 | 0.62 | 158 | 0.101 | 0.68 | 158 | 0.083 | 0.54 |
| 75 | 0.134 | 0.92 | 75 | 0.096 | 0.60 | 159 | 0.013 | 0.03 | 159 | 0.012 | 0.01 |
| 76 | 0.012 | 0.02 | 76 | 0.016 | 0.02 | 160 | 0.012 | 0.02 | 160 | 0.013 | 0.02 |
| 77 | 0.020 | 0.06 | 77 | 0.013 | 0.02 | 161 | 0.015 | 0.04 | 161 | 0.015 | 0.04 |
| 78 | 0.020 | 0.06 | 78 | 0.015 | 0.03 | 162 | 0.053 | 0.31 | 162 | 0.045 | 0.25 |
| 79 | 0.119 | 0.80 | 79 | 0.104 | 0.70 | 163 | 0.016 | 0.04 | 163 | 0.015 | 0.03 |
| 80 | 0.080 | 0.51 | 80 | 0.056 | 0.34 | 164 | 0.067 | 0.41 | 164 | 0.058 | 0.35 |
| 81 | 0.021 | 0.02 | 81 | 0.009 | 0.02 | 165 | 0.014 | 0.02 | 165 | 0.014 | 0.02 |
| 82 | 0.114 | 0.70 | 82 | 0.085 | 0.57 | 166 | 0.081 | 0.53 | 166 | 0.068 | 0.43 |
| 83 | 0.021 | 0.020 | 83 | 0.010 | 0.02 | 167 | 0.082 | 0.55 | 167 | 0.081 | 0.54 |
| 84 | 0.113 | 0.71 | 84 | 0.067 | 0.45 | | | | | | |

The results shown in Table 6 indicate that the formaldehyde emissions measured using the desiccator method at 45 °C for 4 h was almost the same as that under 20 °C for 24 h. Among the 167 groups of data, there were 94 groups with a variation coefficient less than

20%; 56.2% of the data satisfied the preset results, and all the data in group A were greater than those in group B. This was consistent with the variation coefficient of 10%–20% in the desiccator method for the same panel (Myers, 1983) [21]. Therefore, the preset test results were as follows: the formaldehyde emissions measured using the desiccator method under 45 °C for 4 h was basically the same as that under 20 °C for 24 h.

Through a correlation analysis of the 167 groups of data, the formaldehyde emissions measured using the desiccator method at 20 °C for 24 h were taken as the *Y*-axis, and the formaldehyde emissions measured using the desiccator method at 45 °C for 4 h were taken as the *X*-axis. A scatter plot of the corresponding relationship between two different conditions possessed a correlation value of $R^2 = 0.9125$ (Figure 2). Through a correlation analysis using SPSS, the correlation coefficient of the results obtained between the two conditions was R = 0.955, indicating that the correlation was very close. At the same time, a significance test was carried out; the results were extremely significant: sig.(2-tailed) $p < 0.01$. According to the significance test and correlation analysis, a significant linear correlation was observed between this group of experimental data (Table 7).

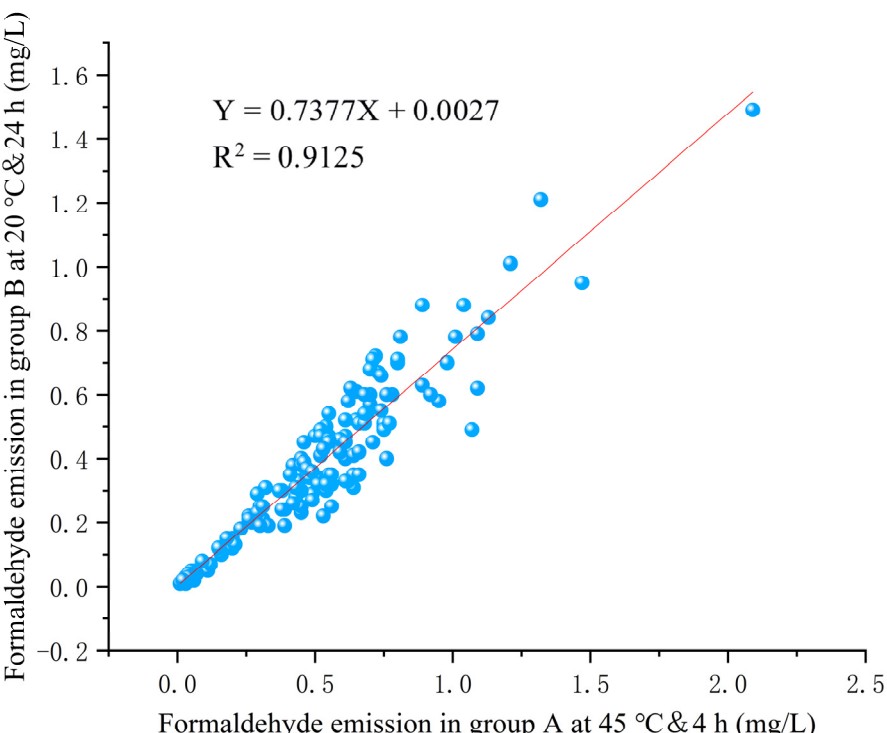

**Figure 2.** Correlation analysis scatter plot of formaldehyde emissions at 45 °C for 4 h and 20 °C for 24 h.

**Table 7.** Correlation analysis of formaldehyde emissions at 45 °C for 4 h and 20 °C for 24 h.

|  |  | 45 °C & 4 h | 20 °C & 24 h |
|---|---|---|---|
| 45 °C & 4 h | Pearson correlation | 1 | 0.955 ** |
|  | Sig. (2-tailed) | / | 0.000 |
|  | Number of cases | 167 | 167 |
| 20 °C & 24 h | Pearson correlation | 0.955 ** | 1 |
|  | Sig. (2-tailed) | 0.000 | / |
|  | Number of cases | 167 | 167 |

** At the 0.01 level (2-tailed), the correlation was significant.

## 4. Conclusions

By comparing the formaldehyde emissions measured using the desiccator method under the test conditions of Chinese national standard at 20 °C for 24 h with that measured under different temperature and time conditions, it was found that the formaldehyde emissions measured using the desiccator under 40 °C for 6 h conditions were close to twice as much as that under 20 °C for 24 h conditions. Under 45 °C for 4 h conditions, the formaldehyde emissions measured using the desiccator were almost equal to that under 20 °C for 24 h conditions. There was a corresponding relationship between the formaldehyde emissions measurement results under these two conditions and those under the standardized conditions, which can be applied in the production control of the furniture industry, theoretically, i.e., if the desiccator measurement under 40 °C for 6 h and 45 °C for 4 h conditions can be used for decisions, which can effectively shorten the formaldehyde measurement time from 24 h to 6 h and 4 h, greatly improving the measurement efficiency. In the case of an emergency, the measurement result can be provided within one working day.

Limitations: This study was trial research carried out in a single enterprise, and the sample was commonly used particleboard of this enterprise, lacking comparison with the test results of other testing institutions. In the future, other enterprises can consider further exploration on this basis and using different sample materials, to improve the approximate equal coincidence rate measured under these two conditions and increase its credibility.

At the same time, there was another insufficiency of this research: In order to find the test conditions that have a certain correlation with the results of the standardized desiccator, multiple test conditions were set, and the experimental data was selected under four groups of temperature and time conditions for comparative analysis. The number of test samples were different under each condition; it was according to the analysis results of the test data to decide whether to continue the test under this condition, when the analysis results of test data conform to the expected trend, and test data was increased as much as possible to obtain more accurate and stable results. The minimum test number was 6 and the maximum number was 169 in this research. To determine how many tests should be carried out to verify the accuracy and how to quantify the accuracy for each range in a shorter time and at a lower cost, further exploration is needed in the follow-up research.

**Author Contributions:** Conceptualization, Z.Z.; methodology, J.Z.; validation, Z.G., data curation, Y.C. and L.X.; writing—original draft preparation, J.Z.; writing—review and editing, F.S. and Z.G.; visualization, Y.C. and Z.G.; funding acquisition, J.Z. All authors have read and agreed to the published version of the manuscript.

**Funding:** This research was funded by Chinese national Promotion Program of Forestry and Grassland Scientific and Technological Achievements, grant number 2020133139.

**Data Availability Statement:** Not applicable.

**Conflicts of Interest:** The authors declare no conflict of interest.

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
