# Peer review of "Faster Measurement for Formaldehyde Emissions from Veneered Particleboard Based on the Standardized Desiccator Method"

_forests, doi:10.3390/f13101566_

Round 1
Reviewer 1 Report (New Reviewer)
The manuscript deals with the investigation and evaluation of correlation between the standardized desiccator method for determining the free formaldehyde release from wood-based composites (particleboards), and the method applied under different temperature and time conditions. In general, the manuscript is well-written and structured, but needs serious improvements before acceptance for publication in the Forests Journal. Please, see below my comments on your work:
Line 2-4: The title of the manuscript should be revised/reformulated to better match the aims and objectives of the manuscript, now it is a bit confusing.
In general, the abstract (lines 12 to 29) and the keywords (lines 31-32) are relevant to scope of the manuscript. I would recommend to add more specific results obtained for the formaldehyde emission (mg/L) from your experimental work.
Introduction: please add some information, supported by relevant references, on the harmful effects of free formaldehyde emission from wood-based composites on the environment and human health.
Please add information and comparison of the formaldehyde emission standards used also in the USA, Europe, Australia, and Japan. Please check this relevant reference on the topic: https://doi.org/10.1080/17480272.2022.2056080
Please extend the information provided on the various factors, affecting the formaldehyde emission from wood composites.
Line 39: please add the standard used (GB18580-2017) in the references of your manuscript.
Lines 48-49: please use the journal numbered format for your references. i.e. [2,3].
Line 59: please provide the full term, i.e. medium-density fiberboard, followed by the common abbreviation MDF.
Line 75: “particle board” should be one word, please revise. “medium- and high-density board”? May be medium-density fiberboard and high-density fiberboard?
Line 94: please add the standard used (GB/T 17657-2013) in the references of your manuscript. The authors have included the standard in the Materials and Methods (line 102), but it should be done at the first place in the manuscript where it was used.
In general, the Introduction part is well-written and informative, and provides relevant background of the research, but has to be further elaborated. The inclusion of additional references, as recommended, is needed to increase the scientific soundness of the work.
Lines 106-107: please provide relevant references to support this statement.
Line 109: please explain the selected temperature and time conditions. How did you select them? Based on your previous trials, experiments by other authors, previous research works?
Line 111: please provide relevant information on the particleboard used for conducting the experiments, e.g. single-layer, three-layer, thickness, density, type of adhesive used, etc.
Overall, the Materials and Methods section is well written but should be further elaborated based on the comments above.
Lines 159-160: “close to the normal living environment temperature” – please rephrase.
Line 187: please add relevant information in the Materials and Methods section on the software used for statistical analyses (SPSS), i.e. company produced, version, etc.
In general, the results of the study are detailed and informative, but are not properly discussed with previous research works in the field. The authors referred to only 1(!) paper – [10]. This is a serious flaw of the presented manuscript and this part should be revised. Please rename this section Results and Discussion, and compare your findings with previously reported works in the field.
In general, the Conclusion part (lines 342-357) reflects the main findings of the manuscript. Please add the also the limitations of your research.
Line 355: please explain what do you mean by “qualified enterprises”
The References cited are appropriate to the topic of the manuscript, but their number (only 13) is absolutely insufficient. The inclusion of additional references, especially in the Introduction and Results sections, will significantly increase the scientific merit of the presented manuscript. In addition, please provide more recent references, now the majority of papers cited are quite old.
Please also refer to the Instructions for Authors for the proper way of formatting your references.
Best regards!
Author Response
Please see the attachment.

Reviewer 2 Report (New Reviewer)
Dear authors,
this work is a method development starting from the standardized desiccator method to have an improved much faster substitution method. The result are clearly presented, yet at some point the conclusion are way too general, aspecially for a person working in industry and lacks some evaluation for the practical sense (although stated that the resulst "can be appied in the actual production control") - in fact for an application some more evaluation of the present data is needed.
Particular remarks:
Engish HAS to be improved. Please check the language with a native speaker. The text is fully mined with non-English sentences and structures.
L116: the test was done according to the standardized GB/T17657-2013 method. Now: is there any guideline for the number of specimens to be tested for a sample? You carried out here 4 set of measurements, once for 53, then 23 then 167 speciemens. What actually does the method say? This is essential for standard deviation and accuracy issues thus for sample comparison. Why did you have different number of specimens in each test, please explain!
L109: Why was 43 and 45 C chosen? These conditions seem to be chosen arbitrarily, and never explained why. Why not 45 and 47 C?
L234-236: IT would be good news, but if I was a company I would ask you how reliable you method is? What is the standard error and accuracy of your method? And finally what is the LOWEST number of specimens I would need to do measurements to have a "preliminary" determine formaldehyde emission? At 40C 6h you have 53 samples, for 45C 4 you had 167!!? In short how can I apply and how reliable is your method? It is OK that you have nice correlations, but in itself it is not enough for practice. See: you points are better fitted (are closer to) the correlation line in Figure 2 for low emission and very high emission values. For midrange emission points are more distant, that means your method is more accurate for very low and very high values but for midvalue results points are very far from the line, thus for these values you method is not so accurate. You should quantify the accuracy for each range before presenting the method for industy. On the other hand I do not find it likely that for each sample they will run 167 specimens just to have a 0.91 R2 in general, it is just too much time and money.
The method otherwise is OK, but should be better described for pratical issues inluding the above detailed statistical properites (std. error, accuracy, std. dev at different ranges and for given number of samples).
Round 2
Reviewer 1 Report (New Reviewer)
Dear authors, thank you for addressing all my previous comments/remarks. I believe the manuscript can be accepted for publication in its current form. However, please try to revise the title of the manuscript. At least, it should be "Faster measurement OF formaldehyde emissions...".
Reviewer 2 Report (New Reviewer)
the paper is well added and the arised concerns have been answered correctly.
This manuscript is a resubmission of an earlier submission. The following is a list of the peer review reports and author responses from that submission.
Round 1
Reviewer 1 Report
I reviewed for Forests the manuscript titled "Rapid Measurement Method for Formaldehyde Emission Based on a Desiccator". While the topic has merit, the paper has significant shortcomings with regard to both originality as well as presentation and interpretation of the results.
Some comments and suggestions:
The manuscript is written as a short communication not as a research article.
Title: The title of the manuscript is not correct since it does not reflect the content of the article. From the title follows that authors developed a new method for the measurement formaldehyde emission from particleboards. However, as follows from the content of the manuscript, the authors changed only the time and temperature conditions of the desiccator method. Therefore, this is the desiccator method for the measurement FE with some changes of measurement conditions.
Introduction: This section is poor written without deep analysis of the studied problem. The authors used only 6 references, a lot of literature resources are missing. The objectives of the study should be clearly described at the end of this section.
Materials and Methods: The first paragraph of this section is not necessary since the authors use standardized desiccator method for the measurement of FE. It is more important to justify the time and temperature conditions of proposed measurement - 60 ℃&6 h, 40 ℃&6 h and 43 ℃&4 h. This information is missing in this section. Why veneered particleboard was chose for the experiment?
Results: The results presented without their deep discussion and their comparison with results obtained by other researches. The statistical analysis of the obtained results is missing.
Conclusions: This section is missing
Reviewer 2 Report
The topic described is important from an industrial and scientific point of view. I want to recommend a few major corrections to improve this paper:
The aim has to be clarified - at the moment, the aim is one long (76 words) sentence, which has to be read a few times to understand.
The introduction has to be more profound. Many good papers explain a correlation between formaldehyde emission and temperature (eg. Zhang, Y., Luo, X., Wang, X., Qian, K., & Zhao, R. (2007). Influence of temperature on formaldehyde emission parameters of dry building materials. Atmospheric Environment, 41(15), 3203-3216.; Zhang, J., Song, F., Tao, J., Zhang, Z., & Shi, S. Q. (2018). Research progress on formaldehyde emission of wood-based panel. International Journal of Polymer Science, 2018.; Kim, S., & Kim, H. J. (2005). Comparison of formaldehyde emission from building finishing materials at various temperatures in under heating system; ONDOL. Indoor Air, 15(5), 317-325.;...). If you prepare a better literature revision (introduction), you will show the scientific gap, which will be explained in this paper.
The methods description is not appropriate. Many experiment details are added to the explanation of the results, e.g. 84-87 or 137-139. The temperature and time parameters should be added to "2. Materials and Methods" paragraph.
The text includes many acronyms that are not explained, e.g., "FE" - in tables; "As" - in tables; 20 ℃&24 h. or 45 ℃&4 h in an abstract or results. Some of them are simple, like Formaldehyde Emission, but should be clarified.
Table/figure 1 presents the same results. Please choose one variant of presenting results. I do not understand what the column description "no." means. Is it repetition or a different variant of the experiment? Above questions and comments can be transferred to tables/figures 2, 4, and 5. Why are the numbers of samples different? Some of the samples in a table are divided into groups, e.g., table 2: the first group are samples no 1 and 2; the next group is sample no 3, next is 4 ... then you present a few samples ina one group "no 7-10"...
The "Discussion" paragraph looks like conclusions. Unfortunately, this part does not include important information that comes from the statistical analysis: is a significant difference between methods, or you will find a simple correlation which gives a possibility to reduce the time of the experiment (according to the aim of the work)?
The paper includes many editing mistakes like not appropriate citations, units etc.